**Data Availability Statement:** Data for the study is available publicly at the CDC BRFSS site:https://www.cdc.gov/brfss/annual_data/annual_data.htm.

# E-cigarette use and respiratory symptoms in residents of the United States: A BRFSS report

**Marcia H. Varella**[1], **Olyn A. Andrade**[2‡], **Sydney M. Shaffer**[2‡], **Grettel Castro**[1‡], **Pura Rodriguez**[1‡], **Noël C. Barengo**[1,3,4], **Juan M. Acuna**[5,6]*

1 Department of Translational Medicine, Herbert Wertheim College of Medicine, Florida International University, Miami, FL, United States of America, 2 American University of Antigua College of Medicine, United States of America, 3 Department of Health Policy and Management, Robert Stempel College of Public Health and Social Work, Florida International University, Miami, FL, United States of America, 4 Department of Public Health, Faculty of Medicine, University of Helsinki, Helsinki, Finland, 5 Department of Epidemiology and Aw 8474000331 R-DISC, Khalifa University, Abu Dhabi, United Arab Emirates, 6 CRUSADA, Robert Stempel College of Public Health and Social Work, Florida International University, Miami, FL, United States of America

☯ These authors contributed equally to this work.
‡ OAA, SMS, GC and PR also contributed equally to this work.
* juan.acuna@ku.ac.ae

## Abstract

### Purpose

E-cigarettes are the most common type of electronic nicotine delivery system in the United States. E-cigarettes contain numerous toxic compounds that has been shown to induce severe structural damage to the airways. The objective of this study is to assess if there is an association between e-cigarette use and respiratory symptoms in adults in the US as reported in the BRFSS.

### Methods

We analyzed data from 18,079 adults, 18–44 years, who participated at the Behavioral Risk Factor Surveillance System (BRFSS) in the year 2017. E-cigarette smoking status was categorized as current everyday user, current some days user, former smoker, and never smoker. The frequency of any respiratory symptoms (cough, phlegm, or shortness of breath) was compared. Unadjusted and adjusted logistic regression analysis were used to calculate odds ratios (OR) and 95% confidence intervals (CI).

### Results

The BRFSS reported prevalence of smoking e-cigarettes was 6%. About 28% of the participants reported any of the respiratory symptoms assessed. The frequency of reported respiratory symptoms was highest among current some days e-cigarette users (45%). After adjusting for selected participant's demographic, socio-economic, and behavioral characteristics, and asthma and COPD status, the odds of reporting respiratory symptoms increased by 49% among those who use e-cigarettes some days (OR 1.49; 95% CI: 1.06–2.11), and by 29% among those who were former users (OR 1.29; 95% CI: 1.07–1.55) compared with

**Funding:** The author(s) received no specific funding for this work.

**Competing interests:** The authors have declared that no competing interests exist.

those who never used e-cigarettes. No statistically significant association was found for those who used e-cigarettes every day (OR 1.41; 95% CI 0.96–2.08).

## Conclusion

E-cigarettes cannot be considered as a safe alternative to aid quitting use of combustible traditional cigarettes. Cohort studies may shed more evidence on the association between e-cigarette use and respiratory diseases.

## Introduction

E-cigarettes use, or vaping, was promoted as a safer alternative to conventional cigarettes and a potential alternative in tobacco smoking cessation efforts [1–3]. It was introduced in the US marketplace in 2007 and since then its use has grown exponentially. Estimates based on the 2016–2017 Behavioral risk Factor Surveillance System (BRFSS) indicate a 4.4% prevalence of e-cigarette use for adults (18 years or older) [4, 5] According to the National Youth Tobacco Survey (NYTS) of 2021, about 2.8% and 11.3% of middle and high school students, respectively reported having used e-cigarette at least once in the past 30 days [6].

The increase in use of e-cigarettes is of concern E-cigarette aerosol contains several toxic chemical constituents. For instance, a systematic review published in 2019 listed 84 hazardous chemical compounds were identified in e-cigarettes. The hazardous effects of those compounds are varied and included cytotoxic, carcinogenic, behavioral, cardiovascular, and respiratory system effects [7, 8]. E-cigarette is also associated with wheeze, chronic cough, phlegm, or bronchitis in children and adults [9–11]. Result of a recently published systematic review estimated that the pooled OR associated with e-cigarette use for asthma was 1.39 (95% confidence interval (CI) 1.28–1.51) and for COPD was 1.49 (95% CI 1.36–1.65) [12].

While growing evidence points to detrimental respiratory effects of e-cigarettes use, just a few comparative studies assessed the effects of e-cigarettes in population based, community dwelling adults in the US [13–15]. Additionally, the impact of e-cigarettes in the respiratory system has been assessed mainly by report of asthma and or COPD, which could have underestimated the magnitude of the respiratory effects caused by e-cigarettes.

In this study, we assessed the association between e-cigarette use and the occurrence of respiratory symptoms (namely cough, phlegm production, or shortness of breath) as reported in the BRFSS, to obtain a more comprehensive understanding of the previous potentially underestimated impact of e-cigarette use in adults in the United States.

## Methods

### Design and setting

We used data from the 2017 BRFSS. Briefly, the BRFSS is an ongoing national yearly cross-sectional survey, originally aimed to identify emerging health problems in order to modify public health programs and policies [16]. It consists of phone-based interviews regarding participants health-related risk behaviors, chronic health conditions, and use of preventive services. The inclusion criteria consisted of individuals between the ages of 18–44 and those living within Arizona, District of Columbia, Florida, Georgia, Minnesota, Nevada, Tennessee, and West Virginia. Only these states in the US collected information on the outcome of interest.

## Study variables

E-cigarette usage (or vaping) was categorized as current everyday e-cigarette smokers, current someday e-cigarette smokers, former smokers, and those that never smoked, according to self-report. Outcome of interest was presence of respiratory symptoms, considered present if the participant reported any of the following: cough, phlegm production occurring daily in the past three months, or shortness of breath hurrying on level ground or when walking up a slight hill or stairs. No report of these three symptoms was considered as absence of the outcome. Covariates assessed included participant's age (assessed as a continuous variable and also categorized as either 18–34 or 35–44 years-old), sex, race (White, Black, Hispanic, and "other"), employment status (employed, unemployed, student, and unable to work), education level (less than high school, high school graduate/General Educational Development exam completion, up to 3 years of college, and greater than 4 years of college), family yearly income (≤15,000, >15,000 to 25,000, >25,000 to 35,000, > 35,000–50,000, and > 50,000), marital status (married and unmarried), health insurance status (insured and uninsured), exercise reported in the past 30 days (yes, no), cigarette smoking status (current, former, never smoker), and history of ever being told to have asthma or chronic obstructive pulmonary disease.

## Statistical analysis

A descriptive analysis was conducted to assess overall sample characteristics and to check for missing data patterns. Subsequently, bivariate analyses were done to further assess for potential confounders in the sample. Lastly, unadjusted and adjusted logistic regression analysis were conducted to estimate crude and adjusted odds ratios and corresponding 95% confidence intervals to explore and control for potential confounding variables and to determine potential interactions (effect modifiers). STATA v15 software was used for all analyses [11]. Further analyses restricted to participants who were never smokers and to those never reporting having asthma and COPD were performed.

This study uses publicly available data from the CDC BRFSS. The IRB of Florida International University considered the present study a "Non-Human Subject Research".

## Results

Of the 22,844 participants who completed the BRFSS questionnaire in the selected states in 2017, 18,079 participants met our study inclusion criteria and were assessed. Table 1 displays the characteristics of participants according to e-cigarette use categories. Overall, never users of e-cigarettes were more frequently older, females, of minority race/ethnicity (Blacks and Hispanics), married, of higher education achievement (> 4 years of college) and higher income compared to all other e-cigarette users categories. Never users also reported less frequently smoking of traditional (combustible) cigarettes. For most comparisons, differences were more striking between never users and every day e-cigarette users. The frequency of asthma and COPD report were highest for those who e-cigarettes use frequency was reported as "some days". The overall frequency of the respiratory symptoms assessed was 27.6%, and it varied according to e-cigarette use categorization: it was highest for those reporting as someday current users and lowest for never users of e-cigarettes (44.6% and 23.3%, respectively, p-value <0.0001 for differences in at least one category) (Table 1).

Table 2 displays the crude and adjusted odds ratios (ORs) for participant's selected characteristics and reported respiratory symptoms. In the unadjusted analyses, compared to the never users, all e-cigarette users had significant increased odds of reporting respiratory symptoms. After controlling for confounding factors (participant's demographic, socio-economic,

**Table 1. Selected characteristics of the sample of participants of the BRFSS according to e-cigarette use.**

| | | E-cigarette User category N (%)* | | | | | | | | p-value |
|---|---|---|---|---|---|---|---|---|---|---|
| | | Current every day | | Current some days | | Former | | Never | | |
| Age (median and IQR) | | 29 | (23–36) | 28 | (22–36) | 30 | (25–36) | 34 | (27–39) | |
| Age (in years, categories) | 18–34 | 315 | (73.1) | 539 | (74.9) | 2868 | (72.3) | 6648 | (59.1) | <0.001 |
| | 35–44 | 134 | (26.9) | 220 | (25.1) | 1406 | (27.7) | 5949 | (40.9) | |
| Sex | Male | 309 | (72.7) | 437 | (60.5) | 2371 | (58.9) | 5574 | (45.0) | 0.002 |
| | Female | 140 | (27.3) | 322 | (39.5) | 1903 | (41.1) | 7019 | (55.0) | |
| Race/ethnicity | White | 359 | (79.4) | 525 | (65.5) | 2980 | (63.8) | 7632 | (48.9) | <0.001 |
| | Black | 13 | (3.2) | 66 | (13.0) | 390 | (11.3) | 1745 | (18.7) | |
| | Hispanic | 34 | (7.2) | 105 | (16.0) | 580 | (17.6) | 2283 | (24.6) | |
| | Other | 43 | (10.2) | 63 | (5.5) | 324 | (7.2) | 937 | (7.7) | |
| Marital Status | Married | 187 | (40.6) | 264 | (35.9) | 1714 | (38.1) | 6767 | (50.6) | <0.001 |
| | Unmarried | 258 | (59.4) | 493 | (64.1) | 2538 | (61.9) | 5752 | (49.4) | |
| Employment Status | Employed | 317 | (72.6) | 466 | (58.2) | 3028 | (69.8) | 9071 | (70.5) | 0.001 |
| | Unemployed | 54 | (10.1) | 139 | (20.3) | 616 | (14.0) | 1821 | (15.5) | |
| | Student | 44 | (12.7) | 96 | (15.8) | 362 | (11.5) | 1018 | (10.7) | |
| | Unable to Work | 29 | (4.6) | 48 | (5.4) | 215 | (4.4) | 473 | (3.0) | |
| Education achievement | Up to high school | 27 | (8.8) | 75 | (16.6) | 345 | (12.4) | 1029 | (12.4) | <0.001 |
| | ≥High school/GED | 172 | (43.2) | 307 | (41.3) | 1341 | (31.9) | 3133 | (27.3) | |
| | ≤3 years of college | 173 | (38.5) | 269 | (33.8) | 1558 | (39.2) | 3469 | (31.8) | |
| | > 4 years of college | 75 | (9.5) | 108 | (8.3) | 1023 | (16.5) | 4933 | (28.5) | |
| Annual family income ($) | <15,000 | 33 | (7.9) | 78 | (13.7) | 364 | (8.9) | 972 | (9.8) | 0.001 |
| | 15,000–25,000 | 84 | (23.1) | 181 | (28.3) | 767 | (22.7) | 1833 | (19.5) | |
| | 25,000–35,000 | 59 | (9.5) | 76 | (11.3) | 470 | (12.3) | 1112 | (11.3) | |
| | 35,000–50,000 | 68 | (20.2) | 89 | (17.3) | 566 | (15.1) | 1539 | (14.3) | |
| | >50,000 | 144 | (39.2) | 194 | (29.3) | 1488 | (41.0) | 5342 | (45.1) | |
| Health insurance status | Insured | 363 | (80.0) | 584 | (71.5) | 3391 | (76.8) | 10379 | (79.0) | 0.017 |
| | Uninsured | 83 | (20.0) | 163 | (28.5) | 857 | (23.2) | 2120 | (21.0) | |
| Exercised in past 30 days | Yes | 337 | (70.5) | 585 | (80.3) | 3208 | (75.6) | 9614 | (75.8) | 0.160 |
| | No | 110 | (29.5) | 173 | (19.7) | 1058 | (24.4) | 2967 | (24.2) | |
| Cigarette smoker status | Current | 133 | (29.4) | 434 | (53.4) | 1838 | (42.2) | 1092 | (7.7) | <0.001 |
| | Former | 225 | (46.2) | 105 | (14.5) | 927 | (20.5) | 6094 | (12.0) | |
| | Never | 86 | (24.4) | 217 | (32.1) | 1486 | (37.3) | 9749 | (80.2) | |
| History of asthma | Yes | 80 | (16.4) | 173 | (24.3) | 836 | (19.8) | 1613 | (12.9) | <0.001 |
| | No | 368 | (83.6) | 580 | (75.7) | 3425 | (80.2) | 10946 | (87.1) | |
| History of COPD | Yes | 19 | (3.5) | 65 | (7.6) | 231 | (4.7) | 292 | (2.5) | <0.001 |
| | No | 429 | (96.5) | 687 | (92.4) | 4026 | (95.3) | 12276 | (97.5) | |
| Report of respiratory symptoms** | Yes | 159 | (32.6) | 320 | (44.6) | 1627 | (36.3) | 2882 | (23.3) | <0.0001 |
| | No | 290 | (67.4) | 439 | (55.4) | 2647 | (63.7) | 9715 | (76.7) | |

*Variables are reported as absolute count and % unless specified. **Symptoms were self-reported and included cough, phlegm production, or shortness of breath. GED-general education diploma; COPD—Chronic Obstructive Pulmonary Disease. P-value corresponding to difference for at least one e-cigarette use categories

behavioral -exercise and combustible cigarette use- asthma and COPD status), the associations were attenuated and significant only for those reporting some days of e-cigarette use and for former users compared to those who never used e-cigarettes (OR 1.49; 95% CI: 1.06–2.11and OR 1.29; 95% CI: 1.07–1.55, respectively). Other variables were also found independently associated with the occurrence of respiratory symptoms; Females, Blacks, those with lower education, being unable to work, family annual income lower than 50,000, being a current smoker,

**Table 2. Associations between E-cigarette use, selected participant's characteristics and occurrence of respiratory symptoms in a sample of adults in the US participating at the BRFSS 2017.**

| Characteristics | | Unadjusted | | Adjusted | |
|---|---|---|---|---|---|
| | | OR (95% CI) | p-value | OR (95% CI) | p-value |
| **E-cigarette Use** | Never | Reference | | Reference | |
| | Current—every day | 1.59 (CI 1.15–2.21) | 0.005 | 1.41 (CI 0.96–2.08) | 0.080 |
| | Current—some days | 2.65 (CI 2.04–3.46) | <0.001 | 1.49 (CI 1.06–2.11) | 0.022 |
| | Former user | 1.88 (CI 1.64–2.16) | <0.001 | 1.29 (CI 1.07–1.55) | 0.009 |
| **Age (Years)** | 35–44 | Reference | | Reference | |
| | 18–34 | 1.04 (CI 0.92–1.17) | 0.581 | 0.97 (CI 0.83–1.14) | 0.754 |
| **Sex** | Female | 1.23 (CI 1.09–1.38) | 0.001 | 1.27 (CI 1.08–1.48) | 0.003 |
| **Race** | White | Reference | | Reference | |
| | Black | 1.19 (CI 1.00–1.41) | 0.054 | 1.26 (CI 1.01–1.59) | 0.043 |
| | Hispanic | 0.86 (CI 0.73–1.01) | 0.066 | 0.87 (CI 0.70–1.07) | 0.187 |
| | Other | 0.89 (CI 0.70–1.12) | 0.316 | 1.07 (CI 0.77–1.47) | 0.699 |
| **Marital** | Married | Reference | | Reference | |
| | Unmarried | 1.36 (CI 1.21–1.54) | <0.001 | 1.09 (CI 0.93–1.28) | 0.262 |
| **Employment Status** | Employed | Reference | | Reference | |
| | Unemployed | 1.36 (CI 1.15–1.60) | <0.001 | 1.17 (CI 0.93–1.46) | 0.179 |
| | Unable to Work | 4.26 (CI 3.27–5.56) | <0.001 | 2.47 (CI 1.76–3.48) | <0.001 |
| **Education achievement** | > 4 years of college | Reference | | Reference | |
| | Less than high school | 2.94 (CI 2.39–3.60) | <0.001 | 1.72 (CI 1.28–2.30) | <0.001 |
| | High school/GED | 2.28 (CI 1.95–2.67) | <0.001 | 1.41 (CI 1.15–1.74) | 0.001 |
| | ≤3 years of college | 2.04 (CI 1.74–2.38) | <0.001 | 1.46 (CI 1.21–1.76) | <0.001 |
| **Annual family income ($)** | >50,000 | Reference | | Reference | |
| | <15,000 | 2.67 (CI 2.13–3.36) | <0.001 | 1.54 (CI 1.14–2.08) | 0.005 |
| | 15,000–25,000 | 2.26 (CI 1.89–2.69) | <0.001 | 1.53 (CI 1.23–1.91) | <0.001 |
| | 25,000–35,000 | 1.61 (CI 1.30–2.00) | <0.001 | 1.27 (CI 0.99–1.62) | 0.060 |
| | 35,000–50,000 | 1.85 (CI 1.50–2.27) | <0.001 | 1.51 (CI 1.21–1.88) | <0.001 |
| **Health insurance status** | Insured | Reference | | Reference | |
| | Uninsured | 1.20 (CI 1.04–1.39) | 0.014 | 0.89 (CI 0.73–1.08) | 0.237 |
| **No exercise in past 30 days** | | 1.59 (CI 1.39–1.82) | <0.001 | 1.41 (CI 1.20–1.67) | <0.001 |
| **Cigarette Smokers status** | Never | Reference | | Reference | |
| | Current | 3.15 (2.72–3.66) | <0.001 | 1.99 (CI 1.62–2.44) | <0.001 |
| | Former | 1.24 (CI 1.04–1.48) | 0.015 | 1.03 (CI 0.83–1.28) | 0.791 |
| **History of Asthma** | | 3.22 (CI 2.76–3.77) | <0.001 | 2.63 (CI 2.18–3.17) | <0.001 |
| **History of COPD** | | 0.22 (CI 0.14–0.34) | <0.001 | 0.42 (CI 0.26–0.69) | 0.001 |

BRFSS- Behavior Risk Factor Surveillance System; OR–odds ratio; CI–confidence interval. Adjusted model included all variables sown in the table

having no exercise reported in the past 30 days, and reporting asthma were associated with higher odds for respiratory symptoms in the adjusted analyses.

To study potential interactions, further analyses were conducted assessing the association between e-cigarette use, and respiratory symptoms occurrence in participants who were never smokers. The absolute number of participants who never smoked combustible cigarettes and were 86, 217, 1486, and 9749 for current every day, current some days, former, and never e-cigarette users, respectively. Lastly, exclusion of participants with prior history of asthma and COPD did not affect the estimates of the associations reported for e-cigarette and respiratory symptoms compared to the analyses where those conditions were adjusted for.

## Discussion

Our data revealed that 2.5% and 4.2% of the sample subjects were every day or some days e-cigarette users, respectively. Overall, respiratory symptoms (cough, phlegm production, or shortness of breath) was reported in approximately 28% of the sample, much higher frequency than those symptoms previously reported [9–11]. Compared to never users, e-cigarette use was associated with up to 50% increased odds of respiratory symptoms; the highest odds were found for those using some day, followed by former users of e-cigarette.

While previous research demonstrated that even short-term exposure to e-cigarettes and their toxic compounds produced cough and phlegm in healthy individuals and individuals with a history of asthma [7], most of the previous cross-sectional and longitudinal studies in US adults that were published, assessed the detrimental effects of e-cigarettes focusing on outcomes such as asthma and COPD occurrence [12, 13, 15, 17, 18]. Our results of for someday e-cigarette users suggesting a 49% increase in odds outcome a composite of cough, phlegm production, or shortness of breath, were consistent in magnitude with the previous studies, despite assessing associations independently of COPD, asthma status, and demographic, socio-economic, and clinical characteristics. For instance, in the meta-analyses reported in the integrative review published in 2021 by Wills et al, the pooled odds of asthma and COPD were 39% (OR = 1.39, 95% CI 1.28–1.51) and 49% (OR1.49, 95% CI = 1.36–1.65) higher respectively, for e-cigarette users compared to never users (12). A less marked increase in the odds of cough or phlegm has been also reported in Chinese adolescents in 2012–2013, with e-cigarette users having 1.28 times higher odds to develop those symptoms compared to non-users (regardless of other combustible smoking habits). (11) Our results suggest that increases symptoms occur and are independent of asthma and COPD diagnosis. We also attempted to further assess for potential gradient in the risk according to the frequency of e-cigarette use and our findings indicate that—while for current everyday users the magnitude of the association seemed like the one found for some day users—results were no longer significant after adjustments. Potential reasons or the lack of a gradient effect include systematic misclassification of frequency of e-cigarette use (e.g., underreporting of every-day users), residual confounding (survey lack details combustible cigarette use and comorbidities), and potential power limitations (about 40% less participants in the everyday users than some days users), and lastly, potential threshold effect (symptoms occurring even at less frequent use). Thus, further studies using larger number of e-cigarette users that collect more detailed and valid information on these factors are needed to better understand whether a dose-response gradient exists for the risk of e-cigarettes use and respiratory symptoms.

E-cigarettes are designed to deliver nicotine throughout the body, without the combustion of tobacco. Through inhalation of water vapor to the lungs, numerous toxic compounds similar to those found within conventional cigarettes are released within the body. Examples of these compounds are aldehydes (formaldehyde and acrolein) and e-liquids (propylene glycol and glycerol) [7, 8, 19]. Overtime, these chemicals cause an elevated mucin concentration and result in failed mucus transport, hallmarks seen in patients of COPD (Chronic Obstructive Pulmonary Disease). The chemicals within e-cigarettes also stimulate neutrophils, resulting in the release of myeloperoxidase (MPO), neutrophil elastase (NE) and other granular proteins. The amount of proteins that are released in e-cigarettes is double to what was described for conventional cigarette smoking, and it causes massive structural damage to airways [20]. These changes could explain potential mechanism that contribute to the common symptoms of chronic cough, phlegm, and shortness of breath observed in e-cigarette users.

This study large sample size and the use of systematically collected and comprehensive information from participants from multiple states contribute to increase study's

generalizability when representing the adult population in the US. Yet, some limitations inherent to the cross-sectional and self-reported nature of the survey should be discussed. We lack detailed information on e-cigarette use characteristics such as the age when e-cigarette use was initiated and the daily frequency of e-cigarette use, as such information was not collected by the BRFSS participants in 2017. Additionally, the timing when e-cigarette use and the symptoms occurred is not clear. The BRFSS survey inquired about cough and phlegm occurring in the three months preceding the survey, while e-cigarettes use status referred to the use at the time of the survey. Therefore, the extent of which potential reverse causality accounts for the association described is unknown, and prospective studies are warranted. Furthermore, given the responses were based on self-report, misclassifications of both exposure to e-cigarettes and occurrence of symptoms are possible. While we do not have evidence to help us to estimate the directionality of the bias, we believe that both the frequency of symptoms and of e-cigarettes would be underreported, thus, most likely biasing the association towards the null hypothesis.

In conclusion, in line of previous findings, our results suggest that e-cigarettes may increase the risk of detrimental respiratory symptoms and thus, cannot be considered as a safe alternative to aid quitting use of combustible traditional cigarettes. Population-based cohort further assessing the potential dose-effect relationship between e-cigarettes use and respiratory symptoms are warranted.

## Acknowledgments

We thank the Behavior Risk Factor Surveillance System (BRFSS) working group and the survey participants for their contribution.

## Author Contributions

**Conceptualization:** Marcia H. Varella, Olyn A. Andrade, Sydney M. Shaffer, Juan M. Acuna.

**Data curation:** Juan M. Acuna.

**Formal analysis:** Marcia H. Varella, Grettel Castro, Pura Rodriguez.

**Investigation:** Marcia H. Varella, Olyn A. Andrade, Sydney M. Shaffer, Grettel Castro, Noël C. Barengo.

**Methodology:** Marcia H. Varella, Olyn A. Andrade, Pura Rodriguez, Noël C. Barengo, Juan M. Acuna.

**Project administration:** Marcia H. Varella, Juan M. Acuna.

**Supervision:** Marcia H. Varella, Grettel Castro, Pura Rodriguez, Noël C. Barengo, Juan M. Acuna.

**Writing – original draft:** Marcia H. Varella, Olyn A. Andrade, Sydney M. Shaffer, Grettel Castro, Pura Rodriguez, Noël C. Barengo, Juan M. Acuna.

**Writing – review & editing:** Marcia H. Varella, Olyn A. Andrade, Sydney M. Shaffer, Grettel Castro, Pura Rodriguez, Noël C. Barengo, Juan M. Acuna.

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
