## [Decision Letter · Decision Letter 0]

22 Aug 2022

PONE-D-22-15106E-cigarette use and Respiratory Symptoms in Residents of the United States: A BRFSS ReportPLOS ONE

Dear Dr. Acuna,

Thank you for submitting your manuscript to PLOS ONE. After careful consideration, we feel that it has merit but does not fully meet PLOS ONE’s publication criteria as it currently stands. Therefore, we invite you to submit a revised version of the manuscript that addresses the points raised during the review process.

At this point we would request you to carefully go through the comments made by the two reviewers and address them satisfactorily. We may review your response and then take a decision on your manuscript accordingly.

We look forward to receiving your revised manuscript.

Kind regards,

Koustubh Panda, M. Tech., Ph.D

Academic Editor

PLOS ONE

Journal Requirements:

Reviewers' comments:

Reviewer's Responses to Questions

**Comments to the Author**

1. Is the manuscript technically sound, and do the data support the conclusions?

Reviewer #1: Partly

Reviewer #2: Yes

2. Has the statistical analysis been performed appropriately and rigorously? 

Reviewer #1: Yes

Reviewer #2: No

3. Have the authors made all data underlying the findings in their manuscript fully available?

Reviewer #1: Yes

Reviewer #2: Yes

4. Is the manuscript presented in an intelligible fashion and written in standard English?

Reviewer #1: Yes

Reviewer #2: Yes

5. Review Comments to the Author

Reviewer #1: 

Introduction:

Line 44: I would hate to imply that e-cigs are a “safer alternative”. I would suggest instead of “considered a safer alternative …” to consider “promoted as a safer alternative …”

Line 48: Please update to the most recent data available (currently 2021 data).

Line 50: It is not at all clear that e-cig use reduces combustible tobacco use.

Line 51: E-cigs contain many toxins and carcinogens, full review of the topic is beyond the scope of this report, however authors should not imply that only a couple of toxins are there. I suggest that the authors look more deeply into this topic and produce a brief succinct sentence that accurately portrays the toxicity.

Line 96: Authors claim “This study does not require IRB clearance”, although this claim is probably correct, it would be best for author to have their IRB declare that it is exempt from review rather than to take it on themselves.

Results: It is interesting that some day e-cig use had greater impact than every day use. Why would that be? If there is causation I would expect a dose-response relation unless there is something else confounding …

Discussion should start with how results address specific aims, how results are biologically plausible, and then how results fit with existing research on the topic. Discussion is not a topic review, nor should it present findings not presented in results. Lines 157-167 seems tangential. I will leave it to the authors to identify and other tangential discussion.

Authors need to explain the lack of finding of dose response in e-cig use on respiratory symptoms. Could certain groups of daily users be motivated to under-report?? Could there be different levels of combustible tobacco use or exposure? Is there something else going on? Authors need to contrast their findings with other research on the same topic. For example, see Xie W et al JAMA Netw Open 2020;3:e2020816 and McConnell R Am J Respir Crit Care Med 2017;195:1043–1049

Conclusion should be clear and focused, what is the take home point? “adds to the growing body of literature” is not a conclusion.

Reviewer #2: 

1) About the E-cigarette usage - how long they use per day? AND how may years they have been using?

This will have an impact. Why was this confounder/variable not considered?

2) What is the reason 'someday current users' have more frequent respiratory symptoms compared to 'everyday current users'?

This has to be explained as well. Much of the discussion has been about 'someday users', ignoring the 'everyday users'.

6. PLOS authors have the option to publish the peer review history of their article (what does this mean?). If published, this will include your full peer review and any attached files.

Reviewer #1: No

Reviewer #2: No

---

## [Author Response · Author response to Decision Letter 0]

19 Oct 2022

Responses to reviewer 1:

Comment#1: Line 44: I would hate to imply that e-cigs are a “safer alternative”. I would suggest instead of “considered a safer alternative …” to consider “promoted as a safer alternative …”

Response#1: Agreed with reviewer and changed accordingly. The new sentence reads now as follows: “E-cigarettes use, or vaping, was promoted as a safer alternative to conventional cigarettes…” 

Comment#2: Line 48: Please update to the most recent data available (currently 2021 data).

Response#2: The reference was updated accordingly. Reference list was updated to reflect the updated information/citation. The new citations is the following 

Gentzke AS, Wang TW, Cornelius M, et al. Tobacco Product Use and Associated Factors Among Middle and High School Students — National Youth Tobacco Survey, United States, 2021. MMWR Surveill Summ 2022;71 (5):1-29.

Comment#3: Line 50: It is not at all clear that e-cig use reduces combustible tobacco use.

Response#3: We agree with the reviewer and have removed that statement. 

Comment#4: Line 51: E-cigs contain many toxins and carcinogens, full review of the topic is beyond the scope of this report, however authors should not imply that only a couple of toxins are there. I suggest that the authors look more deeply into this topic and produce a brief succinct sentence that accurately portrays the toxicity.

Response#4: The statement was adjusted to remove the potential emphasis to the two components listed. The statement now included note from a systematic review that listed 84 components of e-cigarettes that have been linked to hazardous effect. The sentence reads now as follows: “For instance, a systematic review published in 2019 listed 84 hazardous chemical compounds were identified in e-cigarettes. The hazardous effects of those compounds are varied and included cytotoxic, carcinogenic, behavioral, cardiovascular, and respiratory system effects”. 

The following text was removed “…including diacetyl or benzaldehyde, that negatively affect the respiratory system.”

Comment#5: Line 96: Authors claim “This study does not require IRB clearance”, although this claim is probably correct, it would be best for author to have their IRB declare that it is exempt from review rather than to take it on themselves.

Response#5: The sentence was Revised to inform that the IRB of the Florida International University reviewed the proposed study for “Non-Human Subject research determination” and it was considered as Non-Human Subject Research. 

It read now as “This study used publicly available data from the CDC BRFSS. The IRB of Florida International University considered the present study a “Non-Human Subject Research”. 

Of note: The project received the approval by the FIU IRB to be performed as a Non-Human Subject Research. This is not considered an exempt category. Instead, it is considered as research that does not require IRB review and approval.

Comment#6: Results: It is interesting that some day e-cig use had greater impact than every day use. Why would that be? If there is causation I would expect a dose-response relation unless there is something else confounding …Discussion should start with how results address specific aims, how results are biologically plausible, and then how results fit with existing research on the topic. Discussion is not a topic review, nor should it present findings not presented in results. Lines 157-167 seems tangential. I will leave it to the authors to identify and other tangential discussion.

Authors need to explain the lack of finding of dose response in e-cig use on respiratory symptoms. Could certain groups of daily users be motivated to under-report?? Could there be different levels of combustible tobacco use or exposure? Is there something else going on? Authors need to contrast their findings with other research on the same topic. For example, see Xie W et al JAMA Netw Open 2020;3:e2020816 and McConnell R Am J Respir Crit Care Med 2017;195:1043–1049

Response#6: We agree with the reviewer. We expanded the discussion on the potential reasons for lack of findings of a dose-response association (lines 153-165 in the tracked version). Potential reasons including residual confounding due to lack of detailed information available for adjustments, misclassification bias and threshold effect were added. 

The following text was added “Potential reasons or the lack of a gradient effect include systematic misclassification of frequency of e-cigarette use (e.g., underreporting of every-day users), residual confounding (survey lack details regarding combustible cigarette use and comorbidities), and potential power limitations (about 40% less participants in the everyday users than some days users), and lastly, potential threshold effect (symptoms occurring even at less frequent use). Thus, further studies using larger number of e-cigarette users that collect more detailed and valid information on these factors are needed to better understand whether a dose-response gradient exists for the risk of e-cigarettes use and respiratory symptoms. "

In addition, we added the citation by Xie as suggested by the reviewer. Of note the results from Xie et al 2020 were not discussed in further detail as it did not assess e-cigarette’s gradient (the categories some day and everyday users were merged into one) and the outcomes reported were COPD, chronic bronchitis, emphysema, and asthma diagnosis, thus, final endpoints contrasting with the outcomes of the present study (respiratory symptoms) which intended to focus on intermediary outcomes. We believed these intermediate outcomes to be affected at greater magnitude than the COPD and other chronic pulmonary diseases as these later would require longer lag time from expose to development of outcome. This difference in outcome choice was one motivation for doing the present study: to assess if there is a potential underestimation of the magnitude of the health risk of e-cigarette when assessing COPD and Asthma as outcomes. For that same last reason, while we alluded to the consistency of our findings with the studies including the one by MCCornell et al 20217, the discussion focused on highlighting the fact that those studies assessed more “final endpoints” (COPD and asthma) and we were interested in assessing how the magnitude could potentially differ when assessing more intermediate endpoints (Cough, phlegm, shortness of breath). 

Comment#7: Conclusion should be clear and focused, what is the take home point? “adds to the growing body of literature” is not a conclusion.

Response#7: We provided a more directive take home point as suggested and the implication based on the findings/limitations

The conclusion now states: 

“In conclusion, in line of previous findings, our results suggest that e-cigarettes may increase the risk of detrimental respiratory symptoms and thus, cannot be considered as a safe alternative to aid quitting use of combustible traditional cigarettes. Population-based cohort further assessing the potential dose-effect relationship between e-cigarettes use and respiratory symptoms are warranted.”

 

Responses to Reviewer #2: 

Comment#1: About the E-cigarette usage - how long they use per day? AND how may years they have been using? This will have an impact. Why was this confounder/variable not considered?

Response#1: We agree with the reviewer that this is an important limitation. Unfortunately, the BRFSS does not include information on the time e-cigarettes have been used. We added that limitation in the discussion and it reads now as follows:

“We lack detailed information on e-cigarette use characteristics such as the age when e-cigarette use was initiated and the daily frequency of e-cigarette use, as such information was not collected by the BRFSS participants in 2017. Additionally, the timing when e-cigarette use and the symptoms occurred is not clear. The BRFSS survey inquired about cough and phlegm occurring in the three months preceding the survey, while e-cigarettes use status referred to the use at the time of the survey.”

Comment#2: What is the reason 'someday current users' have more frequent respiratory symptoms compared to 'everyday current users'? This has to be explained as well. Much of the discussion has been about 'someday users', ignoring the 'everyday users'.

Response#2: We expanded the discussion on the potential reasons for lack of findings of a dose-response association (lines 153-165 in the tracked version). Potential reasons including residual confounding due to lack of detailed information available for adjustments, misclassification bias and threshold effect were added. The following text was added:

“Potential reasons or the lack of a gradient effect include systematic misclassification of frequency of e-cigarette use (e.g., underreporting of every-day users), residual confounding (survey lack details regarding combustible cigarette use and comorbidities), and potential power limitations(about 40% less participants in the everyday users than some days users), and lastly, potential threshold effect (symptoms occurring even at less frequent use). Thus, further studies using larger number of e-cigarette users that collect more detailed and valid information on these factors are needed to better understand whether a dose-response gradient exists for the risk of e-cigarettes use and respiratory symptoms. "

---

## [Decision Letter · Decision Letter 1]

11 Nov 2022

E-cigarette use and Respiratory Symptoms in Residents of the United States: A BRFSS Report

PONE-D-22-15106R1

Dear Dr. Acuna,

We’re pleased to inform you that your manuscript has been judged scientifically suitable for publication and will be formally accepted for publication once it meets all outstanding technical requirements.

Kind regards,

Koustubh Panda, M. Tech., Ph.D

Academic Editor

PLOS ONE

Additional Editor Comments (optional):

Reviewers' comments:

Reviewer's Responses to Questions

**Comments to the Author**

1. If the authors have adequately addressed your comments raised in a previous round of review and you feel that this manuscript is now acceptable for publication, you may indicate that here to bypass the “Comments to the Author” section, enter your conflict of interest statement in the “Confidential to Editor” section, and submit your "Accept" recommendation.

Reviewer #2: All comments have been addressed

2. Is the manuscript technically sound, and do the data support the conclusions?

Reviewer #2: Yes

3. Has the statistical analysis been performed appropriately and rigorously? 

Reviewer #2: Yes

4. Have the authors made all data underlying the findings in their manuscript fully available?

Reviewer #2: No

5. Is the manuscript presented in an intelligible fashion and written in standard English?

Reviewer #2: Yes

6. Review Comments to the Author

Reviewer #2: (No Response)

7. PLOS authors have the option to publish the peer review history of their article (what does this mean?). If published, this will include your full peer review and any attached files.

Reviewer #2: No

---

## [Editor Report · Acceptance letter]

18 Nov 2022

PONE-D-22-15106R1 

E-cigarette use and Respiratory Symptoms in Residents of the United States: A BRFSS Report 

Dear Dr. Acuna:

I'm pleased to inform you that your manuscript has been deemed suitable for publication in PLOS ONE. Congratulations! Your manuscript is now with our production department. 

Kind regards, 

on behalf of

Professor Koustubh Panda 

Academic Editor

PLOS ONE